

# Coupled exuviae of the Ordovician *Ovalocephalus* (Pliomeridae, Trilobita) in South China and its behavioral implications

Ruiwen Zong

State Key Laboratory of Biogeology and Environmental Geology, China University of Geosciences, Wuhan, Hubei, China

## ABSTRACT

Ecdysis was a vital process during the lives of trilobites. In addition to preserving the morphological changes in trilobite ontogeny, the preservation of its action often captured interesting behavioral information. Abundant exuviae of *Ovalocephalus tetrasulcatus* are preserved in the Ordovician strata in central Hubei, China, and some of them are arranged with two or three together end to end or superimposed. The preserved patterns and burial conditions indicate that these specimens were caused by the active behavior of trilobites. It is speculated that these exuvial clusters were formed by two or three trilobites in line to molt; that is, after one trilobite finished molting, other trilobites molted in front of, behind, or overlying the previously molted shells. This ecdysis strategy is interpreted as related to the postulated herding behavior of some trilobites, representing a behavioral response of the trilobites to choose a nearby safe zone during some risky life activities.

Corresponding author
Ruiwen Zong,
zongruiwen@cug.edu.cn

## INTRODUCTION

The biomineralized or chitinous exoskeleton of arthropods hinders the growth of their bodies; therefore, individuals must shed their old shells many times during growth and development (*Moussian, 2013*; *Daley & Drage, 2016*). Trilobites, as an extinct group of arthropods, also needed to shed their shells as they grew (*Fortey, 2014*). Different trilobites exhibited different molting techniques (*Henningsmoen, 1975*); most trilobites shed their shells through separating the librigenae from the cranidium (*McNamara & Rudkin, 1984*; *Whittington, 1990*), whereas for trilobites with librigenae fused with the cranidium, separation of the cephalon from the thoracopygon usually occurred during molting (*Speyer, 1985*; *Wang & Han, 1997*), and some genera had multiple exuvial modes (*Brandt, 1993*; *Budil & Bruthansová, 2005*). In addition to ecdysis reflecting the ontogenetic development process of trilobites, some exceptionally preserved trilobite specimens also contain behavioral information. For example, some trilobites shed their shells by hiding in empty shells or burrows of other animals (*Davis, Fraaye & Holland, 2001*; *Chatterton, Collins & Ludvigsen, 2003*; *Chatterton & Fortey, 2008*; *Zong, Fan & Gong, 2016*), and even

molted infaunally (*Rustán et al., 2011*), reflecting the hiding behavior of trilobites. Some phacopids may also have exhibited asymmetric behaviors during molting (*Zong & Gong, 2017*). Other trilobites collectively shed their shells, which may have been related to molting–mating behavior (*Speyer & Brett, 1985*; *Speyer, 1990*).

South China is an important area for trilobite fossils, and abundant Ordovician trilobites have been collected from Hubei Province (*Lu, 1975*; *Zhou & Zhen, 2008*). However, there are few reports on exuvial specimens and correlation research. Only photographs of exuvial specimens were attached to the identification of genera and species in the paleontological literature; these exuvial specimens have not been systematically described, their patterns have not been classified and explained, and the behavioral strategy of these trilobites during molting has not been analyzed (*Han & Wang, 2000*). Here, I collected many exuviae of *Ovalocephalus tetrasulcatus* (Phacopida, Pliomeridae) from the Upper Ordovician in central Hubei. Some of the specimens were found with two or three in a line, or preserved partly or even completely overlapping; these patterns are regarded as representing the active behavior of trilobites during molting, and may be related to the herding behavior of some trilobites. They provide new material for understanding exuvial techniques of trilobites, and the behavior of trilobites when molting.

## MATERIALS & METHODS

All specimens were collected from the Upper Ordovician Linhsiang Formation of the Daozimiao section in Jinshan County, central Hubei (GPS: N 30°59′51.49″, E113°06′26.04″) (Fig. 1). The Linhsiang Formation is widely distributed in the middle Yangtze region, and is mainly composed of yellow-green calcareous mudstones with a few siliceous mudstones in the Daozimiao section, but differs from the nodular muddy limestones of the Linhsiang Formation in the type section in Linxiang, Hunan Province. In the Daozimiao section, the Linhsiang Formation is conformably underlain by the muddy limestones of the Upper Ordovician Pagoda Formation and overlain by the graptolite shales of the Upper Ordovician Wufeng Formation. The 1.5-m-thick calcareous mudstones from the top of the Linhsiang Formation (Fig. 1D) yield an abundant and diverse trilobite fauna, including members of the Metagnostidae, Cyclopygidae, Phillipsinellidae, Encrinuridae, Telephinidae, Raphiophoridae, Cheiruridae, Dionididae, Trinucleidae, Asaphidae, Pliomeridae, and Remopleurididae. In addition, the Foliomena fauna (brachiopods) (*Zhan & Jin, 2005*), ostracods, echinoderms, machaeridians, and trace fossils are also found in the same horizon. Based on the trilobite assemblage, the age of the top of the Linhsiang Formation is constrained to the middle Katian (early Ashgill) (*Zhou, Zhou & Xiang, 2016*).

The 1.5-m-thick trilobite-bearing calcareous mudstones are homogeneous, without bedding structures. There are no cross-bedding and graded bedding visible in the longitudinal sections of the rocks, but a small amount of authigenic pyrites is preserved in this interval (Figs. 2A–2D). This evidence suggests the calcareous mudstones formed in a relatively calm and deepwater environment, which is consistent with the results of the characteristics of the brachiopod association found in the same interval (*Zhan & Jin, 2005*).

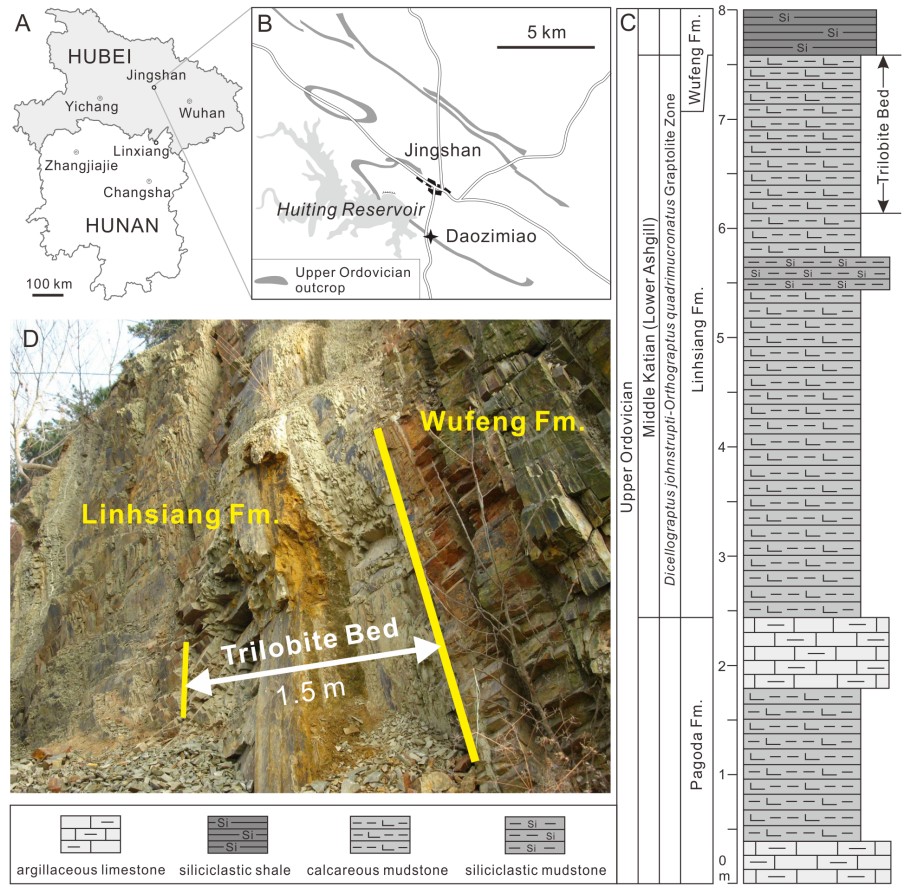

**Figure 1** **Locality map of the fossil site and stratigraphic horizon of *Ovalocephalus tetrasulcatus* in Jingshan County, Hubei.** (A–B) Location of the fossil locality of *Ovalocephalus tetrasulcatus* in the Daozimiao section, Jingshan; (C) stratigraphic column of the Upper Ordovician Linhsiang Formation and stratigraphic distribution of trilobites in the Daozimiao section (graptolite Zone after *Zhou, Zhou & Xiang (2016)*); (D) photo showing the 1.5-m-thick calcareous mudstones from the top of the Linhsiang Formation in the Daozimiao section.

*Ovalocephalus tetrasulcatus* (Kielan 1960) is the only pliomerid trilobite in the Linhsiang Formation of Jingshan. *Ovalocephalus* is largely restricted to peri-Gondwana, but is widely distributed in the Ordovician of China (*Zhou, Yuan & Zhou, 2010*). In the Linhsiang Formation of Jingshan, most specimens of *Ovalocephalus tetrasulcatus* are articulated or nearly articulated exoskeletons, as well as enrolled specimens. This pattern suggests that the trilobites were not transported by current before burial, and were buried in situ. Although some biotic burrows can be observed in the calcareous mudstones (Figs. 2E, 2F), no traces of burrows have been found near the trilobites (Figs. 2B, 2C), which rules out the likelihood that trilobites were affected by biotic disturbance before burial or preserved in burrows.

I collected more than 100 specimens containing articulated or nearly articulated exoskeletons from the calcareous mudstones at the top of the Linhsiang Formation of Jingshan, including 13 specimens that showed two exoskeletons end to end or overlapping one another, and three specimens that show three exoskeletons end to end or two of

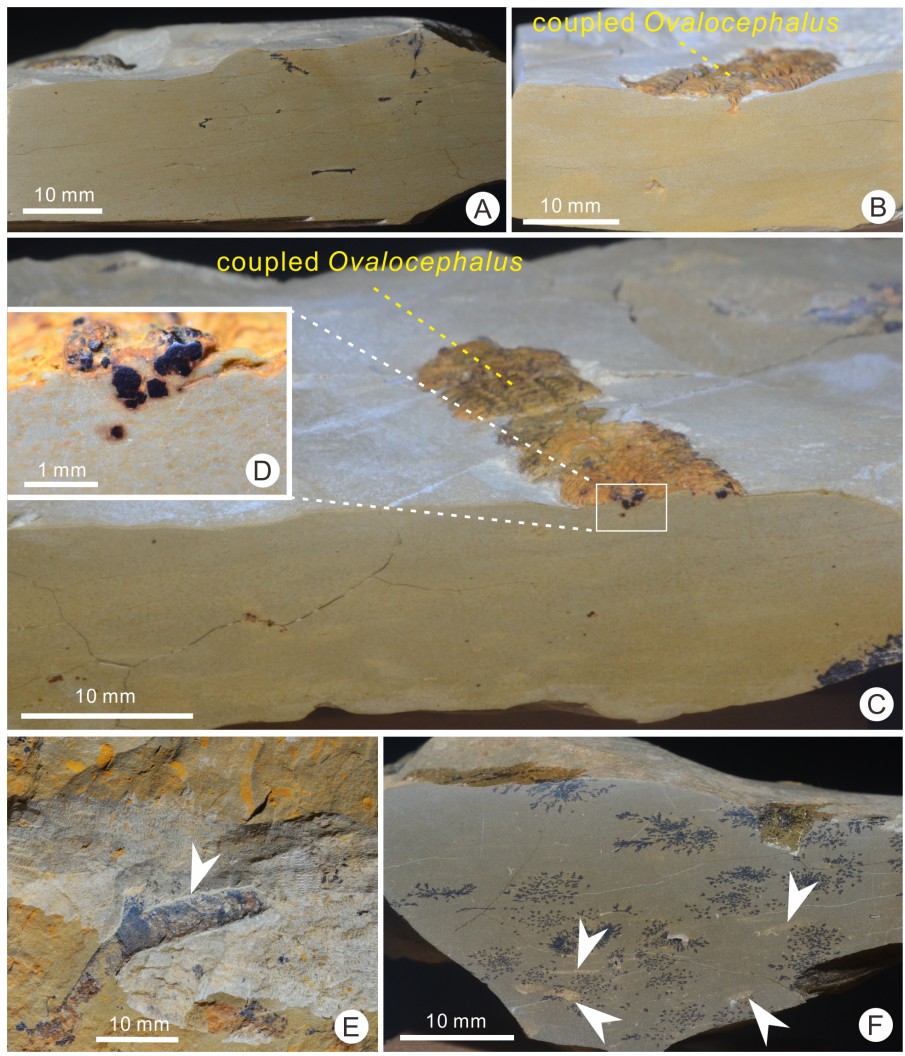

**Figure 2** **Longitudinal sections of the trilobite-bearing calcareous mudstones and trace fossils from the top of the Linhsiang Formation in Jingshan, Hubei.** (A) Longitudinal sections of the calcareous mudstones, without any cross-bedding or graded bedding; (B) longitudinal section of the trilobite-bearing calcareous mudstones; (C) longitudinal section of the trilobite-bearing calcareous mudstones with authigenic pyrites; (D) close up of authigenic pyrites in box in C; (E–F) biotic burrows in the bedding surface (E) and longitudinal section (F) (white arrows) of calcareous mudstones.

them overlapping one another, which obviously differed from isolated single exoskeletons preserved in the same interval. The fossils in Fig. 3 were whitened with magnesium oxide powder, and all photographs were captured using a Nikon D5100 camera with a Micro-Nikkor 55 mm F3.5 lens. The axial azimuth measurements of the trilobites and the rose chart were completed in CorelDRAW X7.

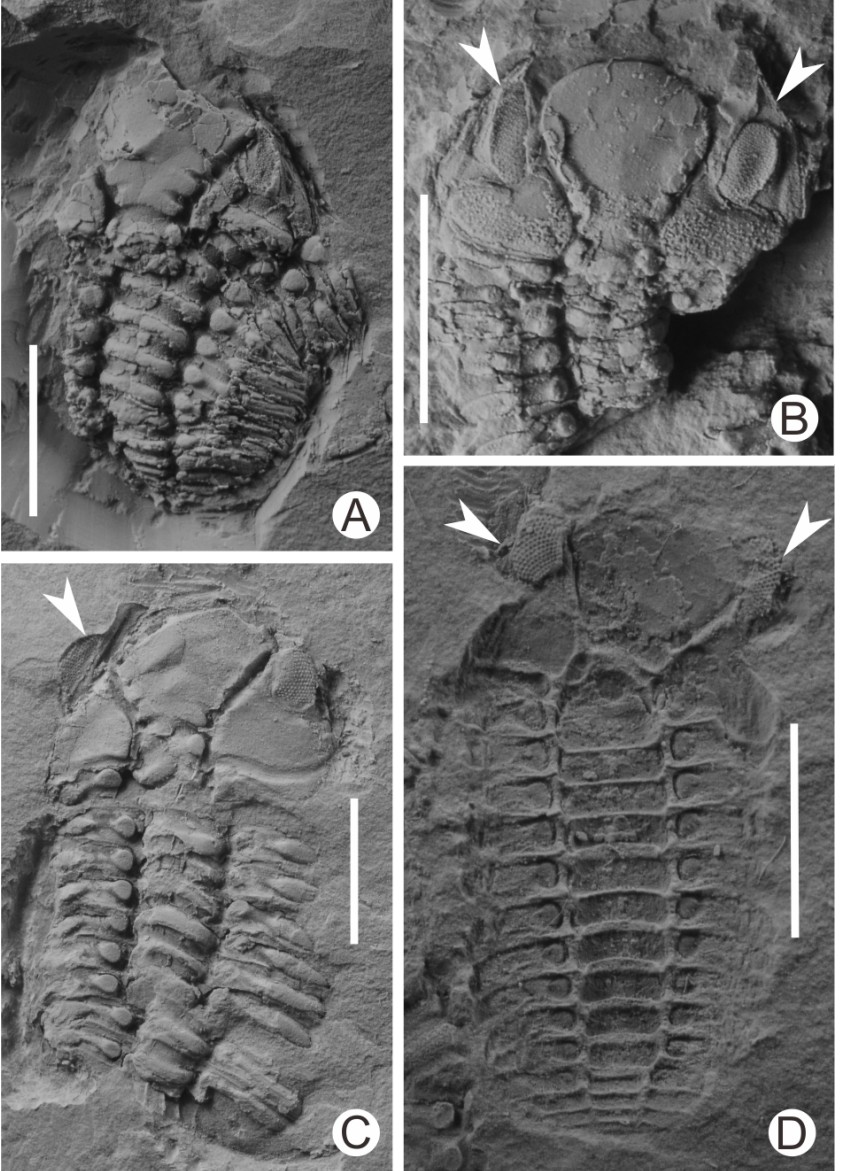

**Figure 3** **Diverse exuvial techniques of *Ovalocephalus tetrasulcatus* from the Upper Ordovician Linhsiang Formation in Jingshan, Hubei.** (A) Cephalon is separated from the enrolled thoracopygon, and overlaps in front of the thorax (CUG-HJ19); (B) two librigenae (white arrows) are separated from the cranidium and preserved nearby (CUG-HJ01); (C) the inverted left librigena (white arrow) is separated from the cranidium, whereas the right librigena is still in situ (CUG-HJ17); (D) two inverted librigenae (white arrows) are separated from the cranidium, which is rotated slightly (external mold) (CUG-HJ20). All scales are 5 mm.

## RESULTS

In the present work, complete exoskeletons include all specimens with articulated cephala, thoraces, and pygidia or enrolled exoskeletons, indicating that they are fossilized carapaces of *Ovalocephalus tetrasulcatus*. Among these nearly complete exoskeletons, a portion of

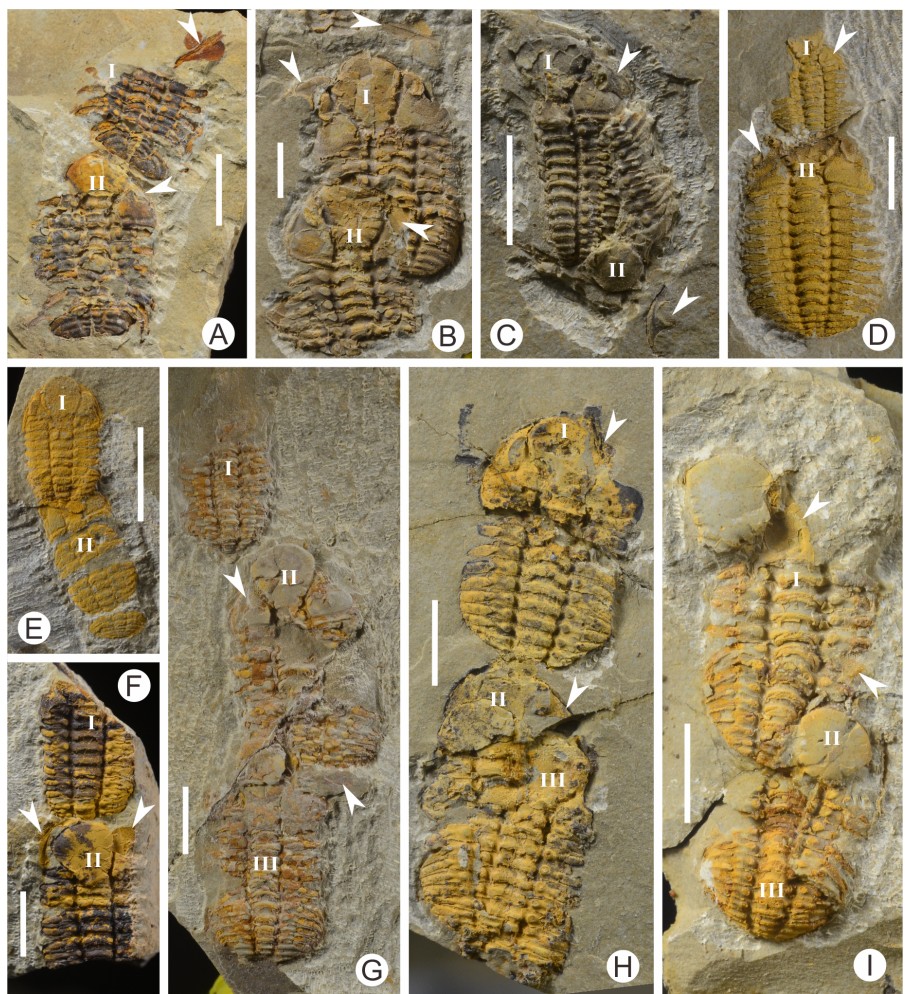

**Figure 4** **Coupled exuviae of *Ovalocephalus tetrasulcatus* from the Upper Ordovician Linhsiang Formation in Jingshan, Hubei.** (A) Two exuviae arranged end to end (CUG-HJ14); (B) two partially overlapping exuviae with the same axial direction (CUG-HJ05); (C) two overlapping exuviae with opposite orientations (CUG-HJ11); (D) two exuviae with different sizes arranged end to end (CUG-HJ02); (E) two small exuviae arranged end to end, with the cephala separated from the thoracopyga, and the first cephalon overlapping with the anterior two thoracic segments (CUG-HJ08); (F) two exuviae arranged in a nearly straight line (CUG-HJ07); (G) three exuviae arranged end to end; the first one is quite different from the other two in size (CUG-HJ03); (H) three exuviae, with the latter two almost completely overlapping and joined with the first one end to end (CUG-HJ06); (I) three exuviae, with the latter two almost completely overlapping and partially overlapping with the first one (CUG-HJ15). The white arrows indicate the separated librigenae, and all scales are 5 mm.

them are articulated thoraces and pygidia, but with the cephala separated and preserved nearby (Fig. 3A), which are considered exuviae (*Han & Wang, 2000*). Another kind has librigenae separated from the cranidia, but the cranidia and thoracopyga are still articulated, or the cranidia are separated from the thoraces and slightly rotated. These specimens include those with two librigenae separated from cranidia, but still preserved nearby (Fig. 3B), and a portion of these librigenae were inverted or rotated (Fig. 3D); in addition, some specimens

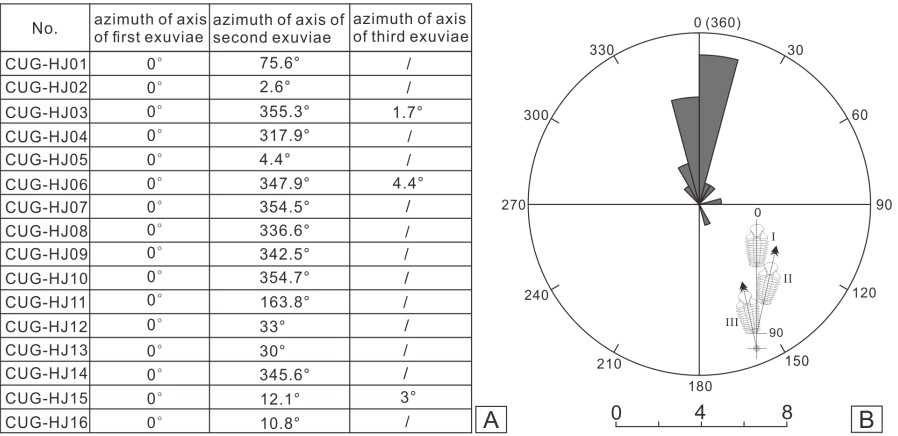

| No. | azimuth of axis of first exuviae | azimuth of axis of second exuviae | azimuth of axis of third exuviae |
|---|---|---|---|
| CUG-HJ01 | 0° | 75.6° | / |
| CUG-HJ02 | 0° | 2.6° | / |
| CUG-HJ03 | 0° | 355.3° | 1.7° |
| CUG-HJ04 | 0° | 317.9° | / |
| CUG-HJ05 | 0° | 4.4° | / |
| CUG-HJ06 | 0° | 347.9° | 4.4° |
| CUG-HJ07 | 0° | 354.5° | / |
| CUG-HJ08 | 0° | 336.6° | / |
| CUG-HJ09 | 0° | 342.5° | / |
| CUG-HJ10 | 0° | 354.7° | / |
| CUG-HJ11 | 0° | 163.8° | / |
| CUG-HJ12 | 0° | 33° | / |
| CUG-HJ13 | 0° | 30° | / |
| CUG-HJ14 | 0° | 345.6° | / |
| CUG-HJ15 | 0° | 12.1° | 3° |
| CUG-HJ16 | 0° | 10.8° | / |

**Figure 5** Statistical results of axial azimuth (A) and the corresponding rose diagram (B) for the coupled exuviae of *Ovalocephalus tetrasulcatus* from the Upper Ordovician Linhsiang Formation in Jingshan, Hubei.

have one librigena separated from the cranidium and inverted, but another still in situ (Fig. 3C). Both types of inverted or rotated librigenae separated from the cranidium are similar to the exuvial mode of many trilobites (*Henningsmoen, 1975*; *McNamara & Rudkin, 1984*), indicating that they are most likely exuviae of *Ovalocephalus tetrasulcatus*, rather than corpses that were broken up by bottom currents and/or organisms, and the natural decomposition of the corpses. This indicates that there is more than one exuvial mode in *Ovalocephalus tetrasulcatus*; i.e., separation of the cephalon from the thoracopygon, separation of the librigenae from the cranidium, or both existing at the same time. A similar diversity of exuvial patterns has been found in other phacopid trilobites (*Brandt, 1993*; *Budil & Bruthansová, 2005*; *Chen, 2011*).

In the sixteen specimens with two or three exoskeletons partly superimposed or end to end, all the specimens show separation of the cephalon or librigenae, and the separated cephala or librigenae are still preserved near the thoracopyga (Fig. 4), indicating that all of these specimens are exuviae of *Ovalocephalus tetrasulcatus*. The central axis of the most frontal *Ovalocephalus tetrasulcatus* in the exuvial clusters was used as the directrix to calculate the axial azimuth of the posterior trilobites. The results showed that there was an obvious dominant orientation; that is, the axial azimuth of the trilobites in these clusters had obvious consistency (Fig. 5), indicating they were not the result of accidental burial events. The separated librigenae preserved near the cranidia may also preclude them from being the result of postmortem transport. Therefore, these exuviae are inferred to have been caused by the active behavior of trilobites.

## DISCUSSION
### Possible causes of coupled exuviae
Preservation of two *Ovalocephalus tetrasulcatus* exoskeletons preserved together is similar to the preservation of an arthropod in the middle of the act of molting (*García-Bellido*

& *Collins, 2004*), but the latter case includes one corpse and one exuvia, whereas all these specimens from Jingshan are exuviae, without corpses; thus, the possibility that the *Ovalocephalus tetrasulcatus* individuals were buried when molting can be excluded. Several cases of queuing of trilobites have been reported in the past, but most of these cases involve corpse fossils, and the number of trilobites is typically more than three; thus, they are regarded as representing unexpected burial during the migration of trilobites (*Radwański, Kin & Radwańska, 2009*; *Błazejowski et al., 2016*; *Vannier et al., 2019*). *Chatterton & Fortey (2008)* reported trilobites aligned for molting, but they were preserved in burrows. In Jingshan, there are no biological burrows near the *Ovalocephalus tetrasulcatus* specimens (Figs. 2B, 2C), and the number of exuviae per specimens is only two or three, in contrast with the former. Since *Speyer & Brett (1985)* first reported synchronous ecdysis in Middle Devonian phacopids in New York, this behavior has been identified in some other trilobites (*Paterson et al., 2007*), usually with many trilobites concentrated in a certain place to shed their shells, which is thought to be related to gregarious behavior and is probably associated with the copulation and reproduction of trilobites (*Speyer, 1990*). A similar pattern may exist in *Ovalocephalus tetrasulcatus*, and pairs of exuviae are relatively easy to understand for mating behavior after molting. However, there are obvious differences of sizes in some exuviae in the same cluster (Fig. 4D), or in different clusters (Figs. 4B, 4E). More importantly, there are also clusters with three exuviae (Figs. 4G–4I), and the superposition (Fig. 4B) even almost complete overlapping (Figs. 4C, 4H–4I) is visible in the exuvial clusters, it is unlikely that two or three trilobites shed their shells synchronously, given the time interval likely represented (a few hours or days). Thus, it is difficult to explain these exuvial clusters as the result of molting–mating behavior.

I speculate that these specimens may reflect a particular exuvial behavior of *Ovalocephalus tetrasulcatus*; because superposition (Figs. 4B–4C, 4H–4I) exist in the exuvial clusters, these trilobites are unlikely to shed their shells synchronously. Instead, they may have shed their shells in lines; that is, after one trilobite finished molting, the others shed their shells as well. Alternatively, perhaps the first trilobite finished molting and left an empty shell, and then later, other trilobites came to the same place to molt. The consistency of the long axes of the exuviae may be related to the seabed topography at that time; for example, molting could have occurred in a narrow shallow gully or on a gentle slope. This is may have been somewhat similar to the molting of extant cicadas on trees, where they occasionally molt in line or overlapping (*Bobo, 2016*; *Muchen, 2016*; *Zhu & Wang, 2017*).

## Implications for the behavioral strategy of trilobites

Arthropods are weak during and after molting, which leaves them vulnerable to predators and even other members of their species. Living shrimps and crabs usually hide in rock crevices or water plants for molting. Trilobites would also have needed a quiet and undisturbed environment when they shed their shells (*Henningsmoen, 1975*; *Han, 2006*). For example, some trilobites used the empty shells of cephalopods and gastropods as shelter for molting (*Chatterton, 1971*; *Brandt, 1993*; *Davis, Fraaye & Holland, 2001*; *Zong, Fan & Gong, 2016*), some even shed their shells under the empty shells of larger trilobites (*Gutiérrez-Marco et al., 2009*), and others molted in burrows of other animals (*Chatterton,*

*Collins & Ludvigsen, 2003*; *Chatterton & Fortey, 2008*). The Upper Ordovician strata in Jingshan have yielded a large number of cephalopods, and I also found nautiloid fossils in the Linhsiang Formation. These predators would have threatened great harm to molting trilobites, and the trilobites therefore would have needed to find safe places for ecdysis. However, when there was no shelter on the seafloor, or insufficient space to hide, possibly, they may have chosen to follow congeneric trilobites nearby to molt; alternatively, the remaining exuviae of other trilobites might have suggested that the location was suitable or safe for molting, thus attracting the later trilobites to molt in the same position, thus forming exuvial clusters. It is worth noting that there are only two or three exuviae in each cluster, which may be because the posterior trilobites adopted the principle of proximity when choosing the molting site. This is similar to the herding behavior of animals in crisis situations, indicating that *Ovalocephalus tetrasulcatus* would take the initiative to choose the nearest safe area to carry out vulnerable life activities.

## CONCLUSIONS

The exuviae of *Ovalocephalus tetrasulcatus* from the Upper Ordovician Linxiang Formation presented two types: separation of the librigenae from the cranidium, and separation of the cephalon from the thoracopygon, reflecting the diversity of the exuvial modes. Some specimens have two or three exuviae arranged end to end, and some have partly and even completely superimposed exuviae together in clusters. The preserved patterns and burial conditions indicate that these specimens are products of the active behavior of trilobites, rather than mechanical transport by currents, unexpected burial in the middle of the act of molting, or collective molting before mating. The preserved patterns and overlapping phenomena of the exuviae indicate that these clusters were formed by two or three trilobites lining up to shed their shells in a long and narrow feature of seafloor topography. They likely represent the behavioral response that the trilobites chose to follow in certain risky life activities, indicating that herding behavior existed in these Ordovician trilobites.

## ACKNOWLEDGEMENTS

I appreciate much the constructive and critical comments from Nigel Hughes and Brian D.E. Chatterton, which aided in the further improvement of the manuscript. I would like to thank Qi Liu from Changsha, Hunan, and Guogang Zhang, Qingsong Gao, and Yang Xu, all from Wuhan, Hubei for their help in the field work.

### Funding

This work was supported by the National Natural Science Foundation of China (41702006) and the Fundamental Research Funds for the Central Universities, China University of Geosciences (Wuhan) (G1323520262). The funders had no role in study design, data collection and analysis, decision to publish, or preparation of the manuscript.

## Grant Disclosures

The following grant information was disclosed by the author:

The National Natural Science Foundation of China: 41702006.

Fundamental Research Funds for the Central Universities, China University of Geosciences (Wuhan): G1323520262.

## Competing Interests

The authors declare there are no competing interests.

## Author Contributions

- Ruiwen Zong conceived and designed the experiments, performed the experiments, analyzed the data, prepared figures and/or tables, authored or reviewed drafts of the paper, and approved the final draft.

## Data Availability

The raw data are photographs in Figs. 1–5.

The specimens are stored in the State Key Laboratory of Biogeology and Environmental Geology, China University of Geosciences, Wuhan, China: CUG-HJ01, CUG-HJ02, CUG-HJ03, CUG-HJ05, CUG-HJ06, CUG-HJ07, CUG-HJ08, CUG-HJ11, CUG-HJ14, CUG-HJ15, CUG-HJ17, CUG-HJ19, CUG-HJ20.

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
