# Peer review of "Coupled exuviae of the Ordovician Ovalocephalus (Pliomeridae, Trilobita) in South China and its behavioral implications"

_PeerJ, doi:10.7717/peerj.10166_

## Round 0.1 · original submission · Major Revisions

This paper has the potential to be an important contribution. However, both reviewers do provide a number of useful suggestions that will help improve it. Please make the changes that are requested, provide a copy of your manuscript showing the changes tracked, and also a letter detailing how you have addressed the comments.

·

Basic reporting

Review of “Coupled exuviae of the Ordovician Ovalocephalus (Pliomeridae, Trilobita) in south China and its behavioral implications (#48931)”

By: Ruiwen Zong

Reviewer: Nigel Hughes

The paper presents data that in the associations of trilobites studied there is a clear tendency for alignment of long axes in clusters of differently sized mature individuals that include several specimens, but rarely more than 4. Many of these individuals show patterns of partial disarticulation, mostly either at the “neck suture” or at the facial sutures. As far as I am aware this particular style of trilobite sclerite association is new and therefore of some interest: it joins a growing literature of case study descriptions that collectively build the descriptive basis of trilobite sclerite disposition. So, in those terms it certainly has merit as a study.

Where I am less keen is the interpretative part. PeerJ is a short format journal, and the paper does not appear to indicate a store of supplementary information that documents and supports the contentions made herein. Such information needs to be available. The paper does refer to some other studies, but it’s unclear whether these provide the specific documentation needed. For example:

1. More information of sedimentology (perhaps on an online supplement?) is necessary. Sedimentary evidence is insufficient to assess represent events taking place in burrows either with or without a roof) or on the seafloor at sediment water interface, and no data is presented on the range of directions that the various clusters pointed towards – only that of the internal variance in orientation within clusters. The sedimentary log provides is fine for orientating us stratigraphically but tells us almost nothing of the details of the depositional regime. The bed yielding the clusters is ~1.5 m thick. What is it comprised of? Are there multiple separate depositional events? What is the relationship between episodes of deposition and the clusters? Here are they located specifically within the bed? Do different clusters on the same bedding surface align? The paper speculates that clusters accumulated on the sediment surface over a period of time. Is there any evidence from sedimentology to corroborate this?

2. As Whittington (1990) pointed out, making the case for trilobite sclerite associations being the product of ecdysis is a challenge. In my view, there is a recent tendency towards tolerating relatively lax criteria for the confident recognition of exuvae. For example, this paper treats the study of Brandt (1993) as an accepted authority – I view that study as flawed in several ways. In my view it’s possible to argue that almost any association of sclerites derived from the same individual is the result of ecdysis – but that doesn’t mean that all actually are the result of that process. As Whittington (1990) made clear, there have to be specific arguments from the disposition of sclerites that make not only ecdysis a candidate process for the disposition, but also one that is more likely than the sum of alternative explanations.

What’s novel (and thus genuinely interesting) here is the association of sclerites derived from individuals and the alignment of multiple individuals, because these particular associations place limits on the processes that can reasonably explain them. What I was hoping for in this paper was, as in Whittington 1990, constraints provided by the alignment pattern itself that independently supports ecdysis as the explanation for the arrangement of the sclerites. I don’t see that here. What if these were animals clustered in burrows that then decayed or were scavenged (see for example Hughes and Cooper, 1999 Journal of Paleontology for an example of scavenging of carcasses)? The pattern of sclerite disposition reported are quite diverse – not the stereotyped associations whose posture can only be reasonably be explained by movements associated with ecdysis. For example, Fig 2 shows a whole range of associations, including a partly enrolled specimen. Of course, the diversity of postures could be because Brandt (and Daley and followers) are right and trilobites did show an unusual diversity of ecdysial strategies atypical compared to living arthropods. But other explanations – such as the disturbance of carcasses (or exuvae for that matter) by various agents (physical processes or biological agents) are possible. How can it be argued that the weight of evidence favors a diversity of modes of ecdysis?

What to say overall? I am very not familiar with PeerJ, but I certainly see a useful publication emerging from this work. It will need careful documentation of sedimentology and taphonomy of the associations, and I’d like to see at least a good portion of that work in the main text of the publication, rather than relegated to the supplementary materials because I think the documentation is critical to interpretation. If PeerJ supports that format then I think a revised version should be encouraged. If not, I would suggest a longer manuscript submitted to a discipline specific journal.

Experimental design

see above

Validity of the findings

see above

Additional comments

see above

·

Basic reporting

Review of “Coupled exuviae of the Ordovician Include” (Pliomeridae, Trilobita) in South China and its behavioural implications by Ruin Song.

1) English language, while clearly not written by an anglophone, is on the whole easy to follow (but see detailed suggested improvements below).
2) The author has done a fairly thorough job of researching the topic, and that is shown by his knowledge of the relevant literature.
3) The figures are of reasonable quality and necessary for the paper.
4) I have a number of suggestions that I hope would serve to improve the paper slightly, and perhaps change the conclusions slightly.
5) The paper is interesting and worthy of publication.
6) Raw data are supplied.
7) The paper includes original research, and improves our understanding of ancient events and biology.
8) Methods are described with sufficient detail and information.
9) On whole, findings are clear, original, but I would suggest that some minor but significant changes should be made to them (see detailed comments below).
10) Manuscript provides interesting new data and ideas.

Detailed comments on MS:
Title: Ovalocaphalus should be in a different font to rest of title to show that it is a binomial name.
Line 16: change “reflecting” to “preserving”.
Lines 21-22: remove “,combined with the overlying phenomenon,” as it is confusing.
Line 24: change “the herding behavior of trilobites” to “the postulated herding behavior of some trilobites”.
Line 25: I do not think that this work has demonstrated what is staled here: “representing a behavioral response of the trilobites to choose a nearby safe zone in a crisis situation”. See more detailed comments below.
Line 27, trilobites do not have an exoskeleton of just chitin, it is mainly calcite with various amounts of Mg, probably with a chitinous outer layer (that is true ion many other arthropods (crabs, lobsters, etc.), so perhaps
Line 35: the thoracopygon is the term used for the thorax and the pygidium (see “morphological terms” in the latest Treatise on trilobites, Whittington et al., 1997, p. 329).
Lines 45-49. Not really relevant to the argument, could be left out.
Line 52: change “literatures” to ‘literature’. Literature is usually used as a collective noun.
Liner 58: change “”related to the herding behavior of trilobites” to :’herding behavior of some trilobites” - since we do not know that all trilobites exhibited this behavior (some probably did not).
Lines 59-60” Change to read “They provide new material for understanding exuvial techniques of trilobites, and the behavior of trilobites when melting”.
Line 70: change “trilobites with the high abundance” to “ an abundant and diverse trilobite fauna”

Line 74: change “machaerids” to “machaeridians”
Line 78-80: Change to read “Ovalocephalus tetrasulcatus (Kielan, 1960) is the only pliomerid trilobite in the Linhsiang Formation. Ovalocephalus is largely restricted to peri-Gondwana, but is widely distributed in the Ordovician of China (Zhou et al., 2010).
Line 87 and 88: Change “together” to “one another”.
Line 89: Change “same horizon” to “ same interval”.
Line 94: Change “In the above materials” to “In the present work”, and change “included” to “include”
Line 95: change “corpses” to “carapaces” or “exoskeletons”. If they are molts it is not appropriate to call them “corpses” as if they were carcasses or cadavers.
Line 102: change “specimens has” to “specimens have”
Line 113” change “thoracopygidia” to “thoracopyga” (see reference above).
Line 124-125: “Remove first “together” and change to read “Preservation of two Ovalocephalus tetrasulcatus exoskeletons preserved together is similar to the preservation of an arthropod in …”
Line 133: either “are no burrows” or “is no burrow”. How about “no evidence of burrows”, as burrows tree often very difficult to see or not preserved in strata that are of a single composition (particularly pure mudstones).
Line 143: change “thus, so, it” to “thus it”
Lines 145-147. I agree that it makes no sense to regard them as the products of a single animal returning to the same place to molt. We do not know how many times trilobites were able to reproduce or whether larger, older trilobites were able to reproduce with younger animals (as occurs in many species today). We do not think that they were like insects and copulated once and then died (although that is a possible scenario for some forms). We also know that trilobites, like any other species probably had a bell curve relative to size (with large and small individuals of the same age and degree of sexual maturity - we certainly have some examples that show a size range in most molt clusters/instars - see my chapter in the treatise and a number of later papers, often with Catherine Cronier as an author).
Line 153: change “latter: to “later”
Lines 156-157: I wonder why you have not considered sea floor currents as a reason for the trilobite molts pointing in the same direction (your effective rose diagram). It makes much more sense that the trilobites were orienting themselves relative to prevailing sea floor currents than a need to invoke linear depressions in an environment where the currents were not strong enough to move the exuviae. The currents could still have been strong enough that a trilobite which was about to lose its exoskeleton (so it would have been slow and weak, and subject to predation) might wish to minimize its exposure to the current (by facing into it to reduce drag forces). I know that at the substrate surface (the base of the hydrodynamic boundary layer) there was practically no current, but unless the trilobite burrowed mostly into the sediment before molting it must have been high enough to have been affected by sea floor currents. We also know that sea floor currents strong enough to form ripple marks can occur at depths of thousands of meters in some areas of the ocean floor (probably not in regions that deposit mudstones?).
Line 163. I suggested that phacopids probably selected nautiloid shells as melting sites as far back as 1971 in Palaeontographica A (for the Devonian of Australia).
Line 167: change “nautilus: to “nautiloid” (could not be Nautilus in the Ordovician!)
Line 186: Why do you think that the trilobites reproduced before molting. If it was at all an active behavior, it would be easier to accomplish while they still had an exoskeleton, and then they could have melted afterwards. Limulus and related forms certainly moulted with the exoskeleton in place, as shown in modern studies and Jurassic trace fossil studies.
Lines 188-190. I am not sure that I like these closing comments. You have not demonstrated that there were any “ crisis situations” unless of course you are referring to attacks by nautiloids. Also, getting together in twos or threes (quite possibly for reproduction” does not really, to my mind, mean “herding behavior”
I have not read through the references carefully, but would point out that on line 249, it is Radwanski and Radwanska, 2009) - I looked at the original paper.
I enjoyed reading this paper.

Experimental design

Ok

Validity of the findings

Slight changes suggested

Additional comments

I enjoyed reading this. I hope that you take my suggested improvements seriously. They are meant to be constructive and to improve the work.

---

## Round 0.2 · Minor Revisions

Please make all of the changes that reviewer Brian Chatterton has requested. Once you have made those changes I will recommend that your paper be accepted. Also, be sure to include with your resubmission a letter describing the changes, as well as a tracked changes version of the manuscript showing the changes that you have made. Thank you.

·

Basic reporting

no comment

Experimental design

no comment

Validity of the findings

The author has done a good job making most of the changes suggested. There are some improvements needed to the English language in the MS (see suggestions in the revised MS sent back. It would have been easier to have made a list of changes that need to be made had there been line numbers in the revised MS.
I am still not convinced why sex could not or would not have been accomplished before molting (which seems to me to be more likely).
The only other general problem that I have with this work is that the author seems to think that only one male is or can be involved in impregnating or fertilizing the eggs of a female, whereas there are numerous examples in the animal kingdom, involving many classes and orders, of more than one male attempting to do this job (ranging from our own species to fishes, squids, bees and probably many other invertebrates, if one was to research this area thoroughly). So the fact that there are three moult carapaces in some of the specimens described in this paper, if they are directly associated with mating behaviour, could simply be the results of there being 2 males and one female in the group.
Just a note that it is the Katian Stage, not the Kaitian.

Additional comments

Please check minor changes to the MS to improve the English language and note the comments above under validity of the findings. Otherwise, I am pleased that you have clearly carefully considered the improvements suggested by the reviewers.

---

## Round 0.3 · Minor Revisions

The paper requires only one minor revision for copy editing/grammatical purposes. With this change it will be ready to go for publication. The necessary change is related to the new text the author inserted to address one of reviewer Chatterton's comments. In particular, for current lines 168-170 it reads "it is unlikely that two or three trilobites shed their shells synchronously, the time interval (such as a few hours or a few days) may be exist in two or three exuvial process. " To correct this, please change the text to "it is unlikely that two or three trilobites shed their shells synchronously, given the time interval likely represented (a few hours or days). "

All of the other changes the author made look good and these addressed all of reviewer Chatterton's comments.

Please make this change and resubmit. Thank you.

---

## Round 0.4 · accepted · Accept

I think this will make a very useful paper and I look forward to seeing it come out.